# Establishment of a Sonotrode Extraction Method and Evaluation of the Antioxidant, Antimicrobial and Anticancer Potential of an Optimized *Vaccinium myrtillus* L. Leaves Extract as Functional Ingredient

**DOI:** 10.3390/foods12081688

**Published:** 2023-04-18

**Authors:** Lidia Gil-Martínez, María José Aznar-Ramos, Maria del Carmen Razola-Diaz, Nuria Mut-Salud, Ana Falcón-Piñeiro, Alberto Baños, Enrique Guillamón, Ana María Gómez-Caravaca, Vito Verardo

**Affiliations:** 1Department of Analytical Chemistry, University of Granada, Avda Fuentenueva, 18071 Granada, Spain; 2Department of Nutrition and Food Science, University of Granada, Campus of Cartuja, 18071 Granada, Spain; 3Biomedical Research Center, Institute of Nutrition and Food Technology ‘José Mataix’, University of Granada, Avda del Conocimiento sn., Armilla, 18100 Granada, Spain; 4Department of Microbiology, University of Granada, Avda Fuentenueva, 18071 Granada, Spain; 5Department of Chemical Engineering, University of Granada, Avda Fuentenueva, 18071 Granada, Spain

**Keywords:** phenolic compounds, bilberry leaves, Box–Behnken design, antimicrobial activity, anticancer activity

## Abstract

*Vaccinium myrtillus* L. (bilberry) leaves are an important by-product of berry production that may be used as a source of phenolic compounds which have a positive effect on human health. Therefore, an ultrasound-assisted extraction via sonotrode has been used for the first time to recover bioactive compounds from bilberry leaves. The extraction has been optimized using a Box–Behnken design. The influence of ethanol:water ratio (*v*/*v*), time of extraction (min) and amplitude (%) were evaluated considering total phenolic content (TPC) and antioxidant capacity (DPPH and FRAP assays) as dependent variables in a response surface methodology (RSM). Optimum values for the independent factors were 30:70 ethanol/water (*v*/*v*), 5 min of extraction and 55% amplitude. The empirical values of the independent variables using the optimized conditions were 217.03 ± 4.92 mg GAE/g d.w. (TPC), 271.13 ± 5.84 mg TE/g d.w. (DPPH) and 312.21 ± 9.30 mg TE/g d.w. (FRAP). The validity of the experimental design was confirmed using ANOVA and the optimal extract was characterized using HPLC-MS. A total of 53 compounds were tentatively identified, of which 22 were found in bilberry leaves for the first time. Among them, chlorogenic acid was the most abundant molecule, representing 53% of the total phenolic compounds identified. Additionally, the antimicrobial and anticancer activities of the optimum extract were tested. Gram-positive bacteria demonstrated high sensitivity to bilberry leaves extract in vitro, with MBC values of 6.25 mg/mL for *Listeria monocytogenes*, *Listeria innocua* and *Enterococcus faecalis*, and 0.8 mg/mL for *Staphylococcus aureus* and *Bacillus cereus*. Furthermore, bilberry leaves extract exerted in vitro antiproliferative activity against HT-29, T-84 and SW-837 colon tumor cells with IC_50_ values of 213.2 ± 2.5, 1140.3 ± 5.2 and 936.5 ± 4.6 μg/mL, respectively. Thus, this rapid ultrasound-assisted extraction method has demonstrated to be an efficient technique to obtain bilberry leaves extract with in vitro antioxidant, antimicrobial and anticancer capacities that may be useful for the food industry as natural preservative or even for the production of functional foods or nutraceuticals.

## 1. Introduction

Bilberry (*Vaccinium myrtillus* L.) is one of the most abundant wild berries in the north of Europe. It belongs to the *Ericaceae* family and its fruits and leaves infusions have been widely consumed through the ages. Berry fruits (and bilberry among them) are nowadays considered “superfoods”, thanks to the health benefits that its high content of bioactive compounds provide [1]. However, less is known about the properties and secondary metabolites present in their leaves, which also might be an interesting source of phytochemicals. Bilberry leaves have higher content of bioactive compounds than fruits [2], and its infusion has been used since ancient times for the treatment of illnesses related to urinary tract, oral inflammation and diabetes. Its content of phenolic compounds, as many research works have proposed, may be involved in antimicrobial, diuretic, astringent and blood glucose-lowering effects [1,3,4].

Phenolic compounds are one of the most studied groups of phytochemicals due to their wide distribution in nature, their high diversity (in terms of chemical structure), biological activity and for being part of the human diet [5]. This large group of molecules ranges from simple phenols to highly polymerized compounds, such as tannins [6]. Their diversity makes them suitable to participate in a wide scope of biological processes. Thanks to their structure, they can act as strong antioxidants exerting beneficial effects in a broad range of pathologies related to oxidative stress and inflammation, including coronary heart disease and cancer [5,7,8]. Polyphenols, such as chlorogenic acid and 3,5-dicaffeoylquinic acid, which are present in bilberry leaves, have demonstrated antiproliferative, anti-inflammatory and antidiabetic activities in numerous studies due to their ability to interfere with proinflammatory cytokines synthesis, immune cell regulation and gene expression [9,10]. Moreover, many phenolic compounds possess significant antimicrobial activity, which may be related to their lipophilia and their interaction with the microbial cell surfaces [11,12,13].

These characteristics make bilberry leaves an interesting source of molecules which may improve health through the fortification of the antioxidant and immune systems. Furthermore, its extract could be used to substitute or potentiate the effect of standard drugs for the prevention or treatment of different diseases [14,15,16].

Therefore, the aim of this study is to optimize a sonotrode ultrasound-assisted extraction of bilberry leaves and to evaluate the bioactivity of the extract in terms of antioxidant capacity, antimicrobial activity and antiproliferative potential. In addition, the identification and quantification of the phenolic compounds of the extract has been carried out to deepen the knowledge of the molecules participating in its biological activity for the promotion and maintenance of health.

## 2. Materials and Methods

### 2.1. Reagents and Plant Material

Folin–Ciocalteu (F-C) reagent, 2,2-diphenyl-1-picrylhydrazyl (DPPH), 2,4,6-tri (2-pyridyl)-1,3,5-triazine (TPTZ), gallic acid and Trolox were purchased from Sigma-Aldrich (St. Louis, MO, USA). Ethanol, water and methanol HPLC grade, glacial acetic acid, iron chloride hexahydrate and sodium carbonate were purchased from VWR (Radnor, PA, USA). Vanillic acid, quercetin, rutin, chlorogenic acid, ferulic acid and catechin standards were also purchased from Sigma-Aldrich (St. Louis, MO, USA). LC-MS grade methanol and water, acetonitrile, sorbic acid (E-200), 5-fluorouracil and sodium acetate were purchased from Merck KgaA (Darmstadt, Germany).

*Vaccinium myrtillus* dry leaves were purchased from Bidah Chaumel (Murcia, Spain). They proceed from an organic field located in Huelva (Spain), and were collected in June 2022. Fresh leaves were air dried in dark conditions at room temperature in order to reach a moisture of 10.5%. They were grounded to an average particle size of 0.8 mm and stored at −20 °C until the extraction.

### 2.2. Extraction of Phenolic Compounds from Bilberry Leaves by Sonotrode

Briefly, 50 mg of grounded *Vaccinium mytillus* dry leaves were extracted with 100 mL of an ethanol/water solution using an UP400St ultrasonic processor (Hielscher, Germany) with the probe S24d14D. Temperature was not monitored. The extracts obtained for each run of the model were centrifuged at 10,000 rpm for 5 min and the supernatants were separated to evaporate the solvent under a rotary vacuum evaporator at 45 °C. Dry extracts were stored at −20 °C until further analysis.

### 2.3. Experimental Design

The optimization of the ultrasound-assisted extraction (UAE) of antioxidant compounds from bilberry leaves was performed using a Box–Behnken design (BBD). It included 15 experimental runs in which the independent variables were ethanol% (*v*/*v*) (X_1_), extraction time (min) (X_2_) and amplitude (%) (X_3_), assayed at three levels (−1, 0, and 1).

Afterwards, total phenolic content (TPC), DPPH and Ferric Reducing Antioxidant Power (FRAP) antioxidant assays were measured in the extracts to be considered as dependent variables in a response surface methodology (RSM) to obtain the optimal conditions for the extraction. In this design, dependent variables were fitted to a second-order polynomial model equation (Equation (1)), where *Y* means the response variable, TPC, DPPH or FRAP values; *Xi* and *Xj* are the independent factors that influence the response; and *β*_0_, *β_i_*, *β_ii_* and *β_ij_* are the regression coefficients of the model (interception, linear, quadratic and interaction terms). The adjustment of the model was evaluated using the analysis of variance (ANOVA) test, taking into account the coefficients, *p*-values and lack-of-fit in the regressions.

Equation (1). Second order polynomial equation for RSM.
(1)$$Y=\beta_{0}+\overset{3}{\underset{i=0}{\sum }}\beta_{i\;}X_{i\;}+\overset{3}{\underset{i=0}{\sum }}\beta_{ii}X_{ii}{}^{2}+\overset{3}{\underset{i=0}{\sum }}\overset{3}{\underset{j=0}{\sum }}\beta_{ii}X_{i}X_{j\;}$$

### 2.4. TPC and Antioxidant Capacity Assays (FRAP and DPPH)

F-C method was assayed to determine the TPC of the extracts [17]. Briefly, 400 μL of sample, and a standard or 80% methanol blank were mixed with 800 μL of 10% (*v*/*v*) F-C reagent in test tubes. Then, 3200 μL of 700 mM Na_2_CO_3_ were added and incubated in the dark at room temperature for 2 h. Finally, the absorbance was measured using a Jenway spectrophotometer (Jenway, Felsted, UK) at 765 nm. Gallic acid was used as a standard and results were expressed as mg gallic acid equivalents/g of dry extract (mg GAE/g d.w.).

FRAP assay was carried out according to the method developed by Benzie and Strain with some modifications [18]. The FRAP reactive solution consisted of ten volumes of 300 mmol/L acetate buffer (pH = 3.6), one volume of 10 mM acid solution of TPTZ and one volume of FeCl_3_ 20 mM. Trolox was used as a standard. A total of 3 mL of FRAP reactive solution were mixed with 480 μL of solvent (blank), and standard or sample in triplicate and the absorbance was read at λ = 593 nm. Results were expressed as mg of Trolox equivalents/g of dry extract (mg TE/g d.w.).

To measure the radical scavenging activity of the extract using DPPH assay, a slightly modified version of the method reported by Brand-Williams et al. [19] was used. Briefly, 10 μL of solvent (blank) or sample were mixed with 190 μL of DPPH 60 μM in triplicate. A calibration curve of Trolox was made to compare the absorbance of the samples after 30 min at 517 nm. Results were expressed as mg TE/g d.w.

### 2.5. HPLC-ESI-TOF-MS Analysis

The analyses were carried out on an ACQUITY UPLC system (Waters Corporation, Milford, MA, USA) coupled with an electrospray ionization (ESI) source operating in negative mode and a time-of-flight (TOF) mass detector (Waters Corporation, Milford, MA, USA). The separation of the compound was achieved through an ACQUITY UPLC BEH Shield RP18 column (1.7 mm, 2.1 mm × 100 mm; Waters Corporation, Milford, MA, USA) at 40 °C, using the gradient and mobile phases described by Martín García et al. [20]. The data were processed in the MassLynx 4.1 software (Waters Corporation, Milford, MA, USA).

### 2.6. Antimicrobial Analysis

Microbial suspensions in saline solution of the targeted microogranisms: *Salmonella enterica* CECT 7160, *Escherichia coli* CECT 405, *Shigella sonnei* CECT 457, *Pseudomonas aeruginosa* CECT 116, *Listeria monocytogenes* CECT 4032, *Listeria innocua* CECT 4030, *Staphylococcus aureus* CECT 239, *Bacillus cereus* CECT 8168, *Zygosaccharomyces bailii* CECT 11997, *Aspergillus niger* CECT 2090 from the Spanish Collection of Type Cultures (CECT) and *Enterococcus faecalis* S-47, *Candida sake* DMC 03 and *Penicillium expansum* DMC 01 (donated by DMC Research Center) were prepared. Mueller–Hinton broth (Scharlau, Barcelona, Spain) was used as a liquid culture medium for bacteria [21] and RPMI-1640 medium with L-glutamine was used for yeast and fungi [22].

Determination of the antimicrobial activity was carried out through the broth microdilution method established by the Clinical & Laboratory Standards Institute (CLSI) [21,22]. Decreasing concentrations of the extract or positive control (E-200) (0.1–50 mg/mL) were prepared in 1:2 dilutions in the corresponding liquid culture medium, which were subsequently inoculated with a microbial suspension to reach a final concentration of 10^5^ colony-forming units/mL (CFU/mL). The obtained dilutions were incubated for 24 h at 37 °C. The MIC was defined as the lowest concentration of the extract that inhibited microbial cell growth as measured by its absorbance at 620 nm. Finally, to determine MBC, samples without cell growth were cultured in agar plates and incubated at 37 °C for 24–48 h to test the presence of viable bacteria. The MBC was determined as the lowest concentration of extract that completely inhibited microbial growth. All assays were performed in triplicate.

### 2.7. Cell Cultures and In Vitro Studies

#### 2.7.1. Cell Lines and Culture

The antiproliferative assays were conducted using a human colorectal adenocarcinoma cell line HT-29 (ECACC 91072201), a human colon carcinoma cell line T-84 (ECACC 88021101) and a human rectum adenocarcinoma SW-837 (ECACC 91031104) that were obtained from the Cell Cultures Unit of the University of Granada (Granada, Spain). Cells were incubated at 37 °C with humidified atmosphere at 5% CO_2_ and cultured with Dulbecco’s modified Eagle medium (DMEM), supplemented with 10% fetal bovine serum (FBS), 10 μL/mL penicillin streptomycin 100× and 2 mM L-glutamine.

#### 2.7.2. In Vitro Antiproliferative Assay

To calculate the half-maximal inhibitory concentration (IC_50_) values of the extract, cells were seeded in sterile 96-well plates (Thermo Fisher Scientific, Roskilde, Denmark) at high density (1.5 × 10^4^ cells/well) and incubated at 37 °C with 5% CO_2_ for 24 h to allow cell adhesion. Increasing concentrations of the extract (31.25–2000 μg/mL) and 5-fluorouracil (5-FU) as positive control (1.95–125 μg/mL) were added in the corresponding wells and were incubated for 48 h at 37 °C with 5% CO_2_. All the concentrations evaluated were performed in sextuplicate. The effect of the extract on tumor colorectal cell lines (HT-29, T-84 and SW-837) was evaluated using the Sulforhodamine-B (SRB) method [23]. Optical density values were determined by colorimetry at 490 nm using a microplate reader (Multiskan EX, Thermo Electron Corporation, Vantaa, Finland). The assessment of absorbance was obtained using the “SkanIt” RE 5.0 for Windows v.2.6 (Thermo Labsystems, Philadelphia, PA, USA) and a mathematical regression analysis for each cell line using a Statgraphics software (Statistical Graphics Corp, 2000, Warrenton, VA, USA) was conducted. The IC_50_ values were calculated from the semi-logarithmic dose–response curve by linear interpolation.

### 2.8. Statistical Analysis

The software, Statistica 7.0 package (StatSoft, Tulsa, OK, USA), was used for experimental design, data analysis and model building. The statistical significance of the model, lack-of-fit and regression terms were evaluated based on ANOVA. The results are expressed as the mean ± standard deviation (SD).

## 3. Results and Discussion

### 3.1. Fitting the Model

The optimization of the sonotrode UAE parameters was achieved through the Box–Behnken experimental design shown in Table 1.

Experimental values of TPC ranged from 130.69 to 181.86 mg GAE/g d.w. of extract. The highest recovery of TPC was achieved using ethanol 55% as solvent and extracting the sample at 25 min at 60% amplitude. The lowest recovery was obtained when ethanol 100% was used as solvent, and the sample was extracted at 5 min at 60% amplitude.

Antioxidant activity values ranged from 73.73 to 253.76 mg TE/g d.w. when measured using DPPH technique and from 114.33 to 295.80 mg TE/g d.w. when using FRAP assay. The maximum values were achieved with 10% ethanol extracted at 5 min at 60% amplitude for DPPH and 55% ethanol extracted at 25 min at 60% amplitude for FRAP antioxidant capacity. Higher extraction times (45 min) at 100% amplitude obtained the lowest antioxidant capacity.

Data shown in Table 1 were used to calculate the combined effect of ethanol/water ratio, extraction time and amplitude on the response variables during the ultrasound-assisted extraction.

The regression coefficients of the model and the analysis of variance results are detailed in Table 2. The model was evaluated according to the significance of the regression coefficients, quadratic correlation coefficients (R^2^) and lack-of-fit. After the ANOVA test, the model was recalculated removing the non-significant terms at a significance level of *p* > 0.05.

The significant interactions on the response variable of TPC were linear ethanol/water (β_1_), linear amplitude (β_3_), crossed ethanol water and time (β_12_), crossed ethanol/water and amplitude (β_13_), crossed time and amplitude (β_23_), quadratic ethanol/water (β_11_), quadratic time (β_22_) and quadratic amplitude (β_33_). For DPPH, all linear effects were significant (β_1_, β_2_ and β_3_). Regarding crossed and quadratic effects, only β_12_, β_23_, β_11_ and β_33_ were significant for DPPH assay. For FRAP assay, all effects were significant (α ≤ 0.05).

The ANOVA test showed the validity of the model, with a significant regression model (*p* < 0.05) and a non-significant lack-of-fit for the three response variables (*p* > 0.05).

### 3.2. Analysis of Response Surfaces

The optimal extraction conditions were determined through the study of the different response surface plots showing the effects of % EtOH (X_1_) with time of extraction (X_2_) and % amplitude (X_3_) (Figure 1 and Figure 2). Each pair of variables was depicted in three-dimensional surface plots, whereas the other variable was kept constant at central level.

The surface response plots obtained for TPC showed that the highest response was obtained at 30–70% amplitude for 2–5 min (Figure 1a), 10–50% amplitude with 20–60% ethanol (Figure 1b) and 10–60% ethanol sonicating for 2–15 min (Figure 1c). For DPPH antioxidant capacity, the highest values were obtained in the range of 40–100% amplitude for 2–10 min (Figure 2a), 10–60% amplitude with 10–50% EtOH (Figure 2b) and 20–50% EtOH for 2–15 min (Figure 2c). In the case of FRAP antioxidant activity, the maximum capacity was obtained for 15–50 min of extraction at 40–100% amplitude (Figure 2d), 20–50% EtOH at 10–60% amplitude (Figure 2e) and 20–50% EtOH for 2–35 min of extraction (Figure 2f).

### 3.3. Optimization of Sonotrode Parameters

The determination for the optimal conditions is the final step after studying the 3-D plots of the RSM. The optimal conditions to obtain the highest content of total phenolic compounds, DPPH and FRAP, of bilberry leaves are summarized in Table 3. The accuracy of the model was established by comparing the predicted values with the empirical results.

The optimal extraction conditions were 30% ethanol/water (*v*/*v*), 5 min of extraction and 55% amplitude. Ethanol demonstrated to be inefficient when used pure or in high concentration, probably due to the solvation provided by the water on the mixture [24]. The optimum extraction time selected was the lowest reporting maximum efficiency on the extraction of antioxidant compounds in order to achieve a rapid environmentally friendly procedure, and with low energy consumption. High times of extraction and amplitudes demonstrated lower antioxidant recoveries, which may be due to the thermolability of the antioxidants present in the raw matrix. The accuracy of the mathematical model was verified through the extraction of bioactive compounds from bilberry leaves using the optimal conditions. The experimental obtained values were not significantly different from the predicted values, showing coefficients of variation lower than 10 for TPC, DPPH and FRAP assays. With regards to the total phenolic content, DPPH and FRAP, empirical values for the optimized extract were 217.03 ± 4.92 mg GAE/g d.w., 271.13 ± 5.84 mg TE/g d.w. and 312.21 ± 9.30 mg TE/g d.w., respectively. Stefanescu et al. evaluated the total phenolic content and antioxidant activity of a bilberry leaves extract using 40% ethanol as solvent in an ultrasonic bath for 30 min., obtaining values of 135.8 mg GAE/g d.w. and 310.74 mmol TE/100 g d.w., respectively [2]. Dragana et al. evaluated the TPC of an aqueous and ethanolic bilberry leaves extracts which obtained the values of 119.17 ± 0.52 and 107.79 ± 1.23 mg GAE/g d.w., repsectively. These results support the higher TPC obtained in this study when increasing the proportion of water in the extraction solvent. Bljajic et al. determined the antioxidant activity of hydroethanolic *V. myrtillus* leaves extract prepared using 80% ethanol and coadjuvated with 30 min of ultrasonication. They obtained TPC values > 400 mg GAE/g d.w. and FRAP values > 450 mgTE/g d.w. [14]. It is important to remark that although other studies have developed bilberry leaves extracts with higher antioxidant or total phenolic content, the second metabolite composition of bilberry leaves is dependent on the geographical location of the plant and environmental conditions. It has been demonstrated that the limiting effect of the temperature on the photosynthesis causes that the leaves of bilberry bushes that grow in high-light intensity locations, higher latitudes and/or high altitudes to have almost two-fold higher concentration of TPC than those that grow in lower altitudes or latitudes [25]. Furthermore, there are seasonal variations on the content and diversity of phenolic compounds and antioxidant activity on bilberry leaves which are dependent on biotic and abiotic stresses. Bujor et al. determined that the best harvest period for bilberry leaves was July or September to obtain the maximum phenolic content recovery [26]. Thus, differences in TPC and antioxidant activity in the final extracts is more related to the geographical origin and harvesting time of the raw materials than the extraction procedure.

### 3.4. Determination of Phenolic Compounds by HPLC-ESI-TOF-MS

Once the extraction method was optimized, the extract was analyzed using HPLC-MS and a total of 53 compounds were tentatively identified. The described compounds, their molecular formula, retention time, experimental and calculated *m*/*z*, score%, and error (ppm) are detailed in Table 4. In order to provide mass accuracy, only molecular formulas with a score higher than 85% were accepted, with an error below 5 ppm. The compounds’ molecular formula was identified by studying and comparing different databases, such as PUBCHEM, Sci-Finder, ChemSpider, literature and by co-elution with commercial standards when available. Vanillic acid was used to quantify hydroxybenzoic acids, ferulic acid was used to quantify hydroxycinnamic acids, quercetin was used to quantify flavonols, rutin was used to quantify rutin and other quercetin derivatives, chlorogenic acid was used to quantify chlorogenic acid and other derivatives, and catechin was used to quantify catechin derivatives and tannins.

A total of 19 phenolic acids and derivatives were identified, which represented 79% of the total phenolic compounds found in bilberry leaves extract. Among them, 3 hydroxybenzoic acid derivatives have not been previously identified in *V. myrtillus* leaves. The first one was identified as dihydroxybenzoic acid pentoside (peak 2), a phenolic compound previously identified in Andean blueberry [27]. The second and third new hydroxybenzoic acid derivatives (peaks 8 and 9) corresponded to two isomers of dihydroxybenzoic acid di-pentoside that were identified by Ammar Sonda et al. in *Ficus carica* leaves [28]. Moreover, 5 derivatives of caffeic acid were detected at retention times 4.419, 4.609, 6.380, 8.333 and 8.999, respectively. Dihydro-caffeoyl-*O*-hexoside (peak 3) was identified by Sun et al. in pear fruits [29], caffeoyl-*O*-hexoside (peak 4) was previously detected in black currant leaves [30] and methyl-5-(6-caffeoyl-glucopyranosyl)-caffeoylquinic acid (peak 15) was characterized in thistle seeds [31]. Caffeoyl hexosyl trihydroxymethoxyphenyl propanoic acid (peak 24) was identified first in blueblerry [27] and subulatin (peak 27) was described for the first time by Tazaki et al. in liverworts [32]. Furthermore, different derivatives of chlorogenic acid, such as chlorogenoyl hexose [27], and dimers of chlorogenic acid were tentatively identified at retention times 4.676, 5.048, 5.118, 5.95, 6.264, respectively. To our knowledge, these derivatives have not been described before in bilberry leaves. These derivatives constituted 23.1% of the total phenolic compounds quantified in the extract. Chlorogenic acid (peak 12) was the most abundant phenolic compound in the extract, constituting 53% of the total phenolic content and 9.1% of the dry extract. These results are in accordance with the work of other authors, who also detected chlorogenic acid as the main phenolic compound in bilberry leaves [33]. Numerous clinical and preclinical studies on the pharmacological effects of chlorogenic acid consumption have been fulfilled. Chlorogenic acid has demonstrated the ability to diminish the risk of various diseases by being involved in immunomodulatory, antioxidant, hepatoprotective, antimicrobial and anticancer processes [34,35]. Therefore, it might play a meaningful role in the maintenance and boost of health. Moreover, two isomers of coumaroylquinic acid eluted at min. 7.054 and 7.278, and coumaric acid malonyl-hexoside was also tentatively characterized at retention time 11.808, which are in accordance with the identification of *V. myrtillus* leaves compounds conducted by Tian et al. [30] and Liu et al. [36], respectively. Feruloylquinic acid was also detected at 7.7 min (peak 21) [2].

Additionally, 20 flavonoids and derivatives were characterized. Flavonoids comprised 21% of the total phenolic compounds of the extract. Most of them have been previously identified in bilberry leaves, such as epigallocatechin (peak 7), quercetin-3-rhamnoside (peak 42), kamepferol-3-*O*-glucoside (peak 34), quercetin rutinoside isomers (peaks 35, and 36), quercetin-galactoside isomers (peaks 37, and 38), quercetin glucuronide (peak 39), quercetin arabinoside (peak 40), quercetin 3-(2″-acetylgalactoside) isomers (peaks 41, and 44), quercetin-3-*O*-rhamnoside (peak 42), quercetin 3-*O*-arabinofuranoside (peak 46), quercetin-HMG-rhamnoside (peak 50) and 3′,7-dimethylquercetin (peak 51) [1,2,5,27,33,34,36,37]. Quercetin glucuronide was the most abundant flavonoid in bilberry leaves, constituting 20% of the total flavonoid content in the extract, which was in accordance with the results obtained by Oszmianski et al. [33]. 6″-*O*-malonylglycitin and isorhamnetin glucuronide (peaks 26 and 45) have been previously characterized in blueberry and blackberry fruits but not in bilberry leaves [27,38]. To our knowledge, it is the first time that quercetin-3-*O*-arabinosylgalactoside, kaempferol 3-*O*-acetyl-glucoside and isorhamnetin-acylated hexoside (peaks 32, 48 and 49) have been identified in this matrix [29].

Furthermore, eight condensed tannins were identified. Five of them were procyanidine type molecules that eluted at retention times 6.723, 7.745, 8.291, 8.572 and 9.863 which were procyanidin dimer, procyanidin-prodelphinidin trimer, procyanidin trimer, procyanidin trimer isomer, and procyanidin dimer isomer, respectively. All of them have already been identified in bilberry leaves [2,30], with the exception of procyanidin-prodelphinidin trimer (peak 22), which was characterized previously by Suvanto et al. in *E. nigrum* [39]. The rest of them were cinchonains (peaks 29, 30 and 33), previously identified in bilberry leaves by Hokkanen et al. [40]. Condensed tannins embodied 16.6% of the total flavonoid content in bilberry extract.

Other newly identified phenolic compounds in bilberry leaves were two isomers of ligustrosic acid (peaks 17 and 20). Moreover, other compounds such as quinic acid (peak 1) and a coumaroyl iridoid, detected at retention time 9.569 min, were identified. Quinic acid was previously characterized in bilberry leaves by Ferlemi et al. [1]. The coumaroyl iridoid (peak 31) has not been previously identified in bilberry leaves but has been identified in blueberry fruits [27].

Results were in accordance with the work of other authors, demonstrating that bilberry leaves is a matrix which exhibits high antioxidant activity that may be related to the presence of phenolic compounds [4,14,26,41].

Bilberry leaves have phenolic acids and derivatives, such as chlorogenic acid, which possess antioxidant activity due to the o-diphenolic functionality and the presence of hydroxyl groups in its moiety. These properties enable the molecule to act as electron and hydrogen atom donor [42]. Flavonoids, such as epigallocatechin and glucosides of quercetin, have the potential to neutralize free radicals, owing to the presence of hydroxyl groups in the molecule and also can act as metal chelating agents [43]. Furthermore, the flavanol class in general and the procyanidin group in particular, exerts the greatest antioxidant capacity among phenolic compounds [44]. There are procyanidin dimers, trimers and cinchonains in the extract which may also contribute to the high antioxidant activity of bilberry leaves extract, owing to the presence of 3′4′-catechol structures in the B-ring coupled with C3-OH and C4-C8 linkages, that strongly inhibits free radicals generation [45].

It is well-known that reactive oxygen species (ROS) and reactive nitrogen species (RNS), such as peroxide, superoxide, hydroxyl and peroxynitrite radicals, generated by cellular processes in the human body are able to react with cellular proteins, lipids and nucleic acids. These reactions generate free radicals that may produce the destabilization of membranes, low-density lipoprotein (LDL) oxidation and damage in DNA structure that may be involved in cellular aging and diseases, such as cancer and coronary heart disease [2,45]. Thus, *Vaccinum myrtillus* L. leaves extract, with its high content of phenolic compounds, may be involved in the prevention of such conditions related to oxidative stress.

### 3.5. Antimicrobial Activity of V. myrtillus L. Leaves Extract

Bilberry leaves extract exerted more antibacterial and antifungal activity as compared to the food preservative sorbic acid (E-200). The results of MIC and MBC/MFC are described in Table 5.

*S. aureus* and *B. cereus* were the most sensitive bacteria to bilberry leaves extract with MIC and MBC of 0.4 and 0.8 mg/mL, respectively, followed by *L. monocytogenes*, *L. innocua* and *E. faecalis* with MIC and MBC values of 3.12 and 6.25 mg/mL, respectively. Bilberry leaves extract revealed antimicrobial activity against the Gram-negatives *S. enterica*, *E. coli*, *S. sonnei* and *P. aeruginosa* at a concentration of 25 mg/mL. The extract exerted antifungal activity against *C.* sake at a concentration of 25 mg/mL and against *Z. bailii*, *P. expansum* and *A. niger* at a concentration of 50 mg/mL. The antimicrobial activity of sorbic acid was also tested as positive control. As it can be seen, the antimicrobial activity of bilberry leaves extract has similar or better values than sorbic acid against Gram-positive bacteria. On the contrary, sorbic acid is more efficient against Gram-negative bacteria and fungi. In general, and according to the results obtained in this paper and in the literature, Gram-positive bacteria are more sensitive to plant extracts than Gram-negatives and fungi [2,46,47]. The higher sensitiveness of Gram-positive bacteria to phenolic compounds may be associated with the hydrophilic protection that outer membrane of Gram-negative bacteria and the cell wall of fungi provides by blocking the interaction of external hydrophobic molecules with the microbial cell surface avoiding its destruction [46].

There are very few research work on the antimicrobial activity of bilberry leaves. Stefanescu et al. evaluated the antimicrobial activity of the extract agaonst *S. aureus*, *E. faecalis*, *R. equi*, *E. coli*, *K. pneumonia*, *P. aeruginosa*, *C. albicans*, *C. zeylanoides* and *C. parapsilosis*. They observed higher activity against *S. aureus*, *E. faecalis* and *R. equi* (Gram-positives) than the other microorganisms tested, which is in agreement with our study [2]. Tian et al. studied bilberry leaves extract growth inhibition against *E. coli*, *S. aureus*, *L. monocytogenes*, *B. cereus* and *S. enterica*. The best results were obtained for *S. aureus*, *L. monocytogenes* and *S. enterica* [46]. Moreover, Dragana et al. studied the antimicrobial activity of *V. myrtillus* leaves extract against human pathogens related to urinary tract infections. They found that *E. coli*, *E. faecalis* and *P. vulgaris* were sensitive to the bioactive compounds in bilberry fruits and leaves [47]. Thus, bilberry leaves extract can be considered a source of antimicrobial molecules that may be used in the food industry as preservatives or for the development of functional food or nutraceuticals.

### 3.6. Antitumor Activity of V. myrtillus L. Leaves Extract

The anticancer activity of bilberry leaves extract was tested against tumoral colon cells HT-29, T-84 and SW-837. Results are shown in Table 6 as IC_50_ values. To our knowledge, this is the first paper in which in vitro antiproliferative activity of *V. myrtillus* leaves has been assayed but some researchers have reported the in vitro antitumoral activity of *V. myrtillus* fruits.

Bilberry extract presented in vitro antiproliferative activity against all the tested colorectal carcinoma cell lines with IC_50_ values of 213.2 ± 2.5 μg/mL in HT-29 cells, 1140.3 ± 5.2 μg/mL in T-84 cells and 936.5 ± 4.6 μg/mL in SW-837 cells. Tumoral colon cells were preferred in this study because from a biological point of view, it is more likely that bioactive compounds have direct contact with tumor cells during the digestion process. Katsube et al. studied the inhibition of the growth of human colon carcinoma HCT116 by berry extracts, wherein the bilberry ethanolic extract was found to be the most effective. They reported that at a concentration of 2 mg/mL, it decreased the number of viable cells by 66% after 48 h of incubation [48]. Wu et al. evaluated the cell growth inhibition and induction of apoptosis in HT-29 cells produced by various berry fruits extracts. Results revealed that bilberry extract was the most significant, reducing 30% of cell growth when tested at a concentration of 10 mg/mL. Moreover, concentrations of 20–60 mg/mL demonstrated to induce apoptosis in HT-29 cells [49]. Savikin et al. studied cell viability of LS147 colon tumor cells when treated with bilberry decoction and bilberry infusion teas, showing IC_50_ values of 176.32 μg/mL and 178.52 μg/mL, respectively [50]. Minker et al. obtained optimal results in assays conducted with two human colon cancer cell lines, which were induced with proanthocyanidins extracted from various berries. Specifically, the EC_50_ (effective concentration, 50%) of bilberry was 24.7 μg/mL in SW620 cells after 24 h and 25.2 μg/mL in SW840 cells after 48 h of induction [51]. Furthermore, Mudd et al. studied the effect of bilberry anthocyanin extract on the proliferation of human colon tumor cells HCT116 and HT-29, and normal colon cells CCD-18Co. They found that both tumoral cell lines were inhibited by the extract at concentrations of 124 μmol/L and 75 μmol/L, but normal cells were only inhibited at concentrations above 1050 mmol/L, so it may suggest that anthocyanins target tumoral over normal colon cells [52]. Mechikova et al. evaluated the antitumor activity of leaves and phytochemicals of *Vaccinium smallii* and found that the cancer-preventive properties of the plant may be related to its content of chlorogenic acid, quercetin and some glycosides of quercetin (mainly, rhamnosyl and xylosyl derivatives) [53]. Proantocyanidins are quite interesting phytochemicals for the prevention and treatment of colon cancer as due to the size of their molecule, they cannot be absorbed in the intestine and can reach the colon to locally exert their activity on tumoral cells [51], although the antitumor activity of plant extracts may be caused by both, phenolic or non-phenolic phytochemicals [49]. Nevertheless, intestinal microbiota may produce changes in the molecular structure of the phytochemicals through fermentation, which might result in the modification of the in situ antitumoral performance of the extract [54]. Thus, more studies are needed to confirm these preliminary results.

## 4. Conclusions

A sonotrode ultrasound-assisted extraction method for the isolation of phenolic compounds from bilberry leaves has been optimized through a Box–Behnken design for the first time. The method developed is fast (the optimal extraction time is 5 min) and environmentally friendly, as it consumes very low energy (optimum amplitude was 55%), and the best extraction solvent is a mixture of ethanol-water (30% *v*/*v*) considered GRAS (generally recognized as safe) and it promotes the revalorization of an agroindustrial by-product, supporting the circular economy. Furthermore, this methodology is suitable to be scaled-up to pilot and industrial scale. Additionally, the comprehensive tentative characterization and quantification of the optimal extract has allowed determining 53 bioactive molecules, of which 22 are described for the first time in bilberry leaves. Among the phenolic compounds identified, there were phenolic acids and derivatives, flavonoids and procyanidins. However, chlorogenic acid stood out, representing 9.1% of the dry extract. Moreover, the bioactivity of the extract in terms of antioxidant, antimicrobial and anticancer capacities have been tested in vitro. Bilberry leaves extract has demonstrated to exert remarkable antioxidant activity, to be able to inhibit the proliferation of Gram-positive pathogenic bacteria, and, have cytotoxic properties for some tumoral colon cell lines, such as HT-29, T-84 and SW-837. Thus, the developed bilberry leaves extract, can be a potential ingredient for the food or pharmaceutical industries as it may be used as a functional ingredient for the development of nutraceuticals, food preservatives or even functional foods. However, further in vitro and in vivo tests should be performed in order to substantiate these results.

## Figures and Tables

**Figure 1 foods-12-01688-f001:**
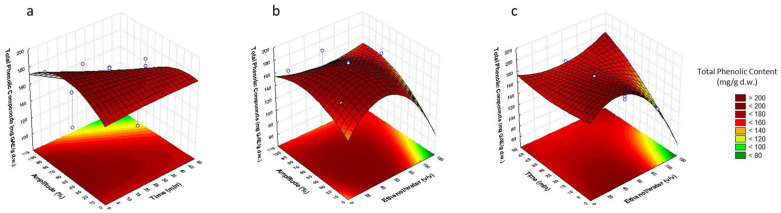
Response surface plots showing combined effects of process variables for TPC (mg GAE/g d.w.): amplitude (%)—time (min) (**a**), amplitude (%)—% EtOH (**b**) and % EtOH—time (min) (**c**).

**Figure 2 foods-12-01688-f002:**
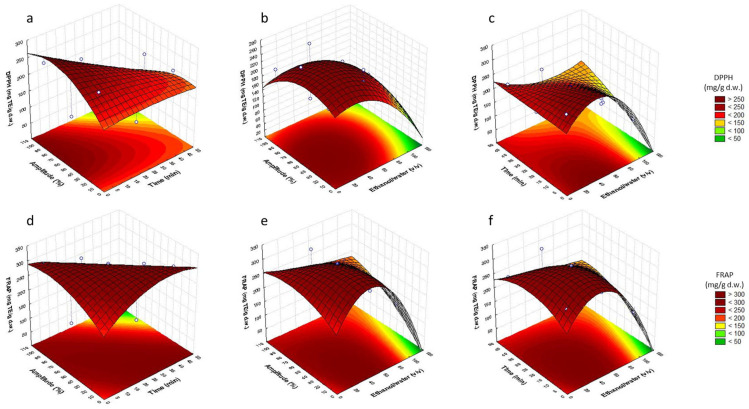
Response surface plots showing combined effects of process variables for DPPH and FRAP (mg TE/g d.w.): amplitude—time (min) (**a**,**d**), amplitude—% EtOH (**b**,**e**) and % EtOH—time (min) (**c**,**f**).

**Table 1 foods-12-01688-t001:** Box–Behnken experimental design including the natural and coded (in parenthesis) values of the extraction conditions and the experimental results for TPC, DPPH and FRAP.

Independent Factors				Dependent Factors		
No	X_1_	X_2_	X_3_	TPC (mg GAE/g d.w.)	DPPH (mg TE/g d.w.)	FRAP (mg TE/g d.w.)
1	10 (−1)	5 (−1)	60 (0)	171.82	253.76	267.28
2	100 (1)	5 (−1)	60 (0)	130.69	126.96	148.01
3	10 (−1)	45 (1)	60 (0)	164.93	218.73	243.48
4	100 (1)	45 (1)	60 (0)	173.61	151.99	184.86
5	10 (−1)	25 (0)	20 (−1)	170.77	190.50	272.97
6	100 (1)	25 (0)	20 (−1)	135.15	116.10	164.67
7	10 (−1)	25 (0)	100 (1)	165.60	200.43	257.46
8	100 (1)	25 (0)	100 (1)	145.44	142.78	195.94
9	55 (0)	5 (−1)	20 (−1)	179.51	253.43	271.62
10	55 (0)	45 (1)	20 (−1)	175.30	220.19	291.60
11	55 (0)	5 (−1)	100 (1)	174.69	234.33	289.50
12	55 (0)	45 (1)	100 (1)	119.82	73.73	114.33
13	55 (0)	25 (0)	60 (0)	181.86	220.20	295.80
14	55 (0)	25 (0)	60 (0)	179.02	224.76	291.70
15	55 (0)	25 (0)	60 (0)	180.80	222.77	288.05

X_1–3_: Ethanol/water (*v*/*v*), time (min.) and amplitude (%). TPC: Total phenolic content. GAE: Gallic acid equivalents. TE: Trolox equivalents. d.w.: dry weight.

**Table 2 foods-12-01688-t002:** Estimated regression coefficients of the adjusted second-order polynomial equation (Equation (1)) and analysis of variance (ANOVA) of the model.

Regression Coefficients	TPC (mg GAE/g d.w.)		DPPH (mg TE/g d.w.)		FRAP (mg TE/g d.w.)	
	Effect	*p*-Value	Effect	*p*-Value	Effect	*p*-Value
β_0_	158.9447	0.0000 *	181.9114	0.0000 *	225.1434	0.0000 *
Linear						
β_1_	−20.1117	0.0028 *	−86.5233	0.0004 *	−87.6007	0.0011 *
β_2_	2.1615	0.1814	−35.6410	0.0023 *	−21.5165	0.0176 *
β_3_	−8.3473	0.0161 *	−15.3924	0.0120 *	−21.3091	0.0179 *
Crossed						
β_12_	24.9041	0.0033 *	30.0333	0.0057 *	30.3200	0.0160 *
β_13_	7.7287	0.0329 *	8.3744	0.0669	23.3861	0.0264 *
β_23_	−25.3309	0.0032 *	−63.6782	0.0013 *	−97.5754	0.0016 *
Quadratic						
β_11_	14.1936	0.0028 *	33.8420	0.0012 *	49.9700	0.0016 *
β_22_	6.1028	0.0147 *	0.8747	0.5381	30.9723	0.0042 *
β_33_	12.1282	0.0038 *	26.2820	0.0020 *	19.1176	0.0110 *
R^2^	0.9978		0.9980		0.9992	
*p* Model	0.0018 *		0.0065 *		0.0024 *	
*p* Lack of fit	0.1814		0.1249		0.5379	

* Significant at α ≤ 0.05; 1 Ethanol-water ratio (*v*/*v*), 2 time, 3 amplitude.

**Table 3 foods-12-01688-t003:** Optimal conditions of extraction and predicted and empirical values of the model are expressed as the mean ± SD (*n* = 3).

Parameter	Optimal Conditions		
Ethanol (%)	30		
Time (min)	5		
Amplitude (%)	55		
	**TPC**	**DPPH**	**FRAP**
Predicted value (mg/g d.w.)	195.59 ± 6.76	276.85 ± 10.98	301.55 ± 18.96
Empirical value (mg/g d.w.)	217.03 ± 4.92	271.13 ± 5.84	312.21 ± 9.30
Coefficient of variation (%)	7.35	1.48	2.46

**Table 4 foods-12-01688-t004:** Characterization and quantification of phenolic compounds using HPLC-ESI-TOF-MS in the optimal extract of bilberry leaves. Results of quantification are expressed as the mean ± SD (*n* = 3).

Peak No.	Retention Time (min)	*m*/*z* Exp.	*m*/*z* Calc.	Molecular Formula	Error (ppm)	Score	Proposed Compound	Quantification (mg/g d.w.)
Phenolic acids and derivatives								
2	3.61	285.0608	285.0610	C_12_H_14_O_8_	−0.7	99.26	Dihydroxybenzoic acid pentose	0.30 ± 0.03
3	4.42	343.1034	343.1029	C_15_H_20_O_9_	1.5	99.98	Dihydro-caffeoyl-*O*-hexoside	1.01 ± 0.07
4	4.61	341.0869	341.0873	C_15_H_18_O_9_	−1.2	93.96	Caffeoyl-*O*-hexoside	<LOQ
5	4.68	515.1406	515.1401	C_22_H_28_O_14_	1	96.08	Chlorogenoyl hexose	0.51 ± 0.04
6	4.96	433.0981	433.0982	C_17_H_22_O_13_	−0.2	99.21	Gallic acid di-pentoside I	<LOQ
8	5.02	417.1037	417.1033	C_17_H_22_O_12_	1	99.99	Dihydroxybenzoic acid di-pentoside isomer a	<LOQ
9	5.05	417.1034	417.1033	C_17_H_22_O_12_	−0.2	99.95	Dihydroxybenzoic acid di-pentoside isomer b	<LOQ
10	5.12	707.1827	707.1823	C_32_H_36_O_18_	0.6	99.20	Chlorogenic acid dimer isomer a	2.54 ± 0.07
11	5.25	707.1826	707.1823	C_32_H_36_O_19_	0.4	93.54	Chlorogenic acid dimer isomer b	7.49 ± 0.14
12	5.68	353.0862	353.0873	C_16_H_18_O_9_	−3.1	100	Chlorogenic acid *	90.66 ± 0.40
13	5.95	707.1813	707.1823	C_32_H_36_O_18_	−1.4	95.65	Chlorogenic acid dimer isomer c	14.20 ± 0.24
14	6.26	707.1829	707.1823	C_32_H_36_O_18_	0.9	99.90	Chlorogenic acid dimer isomer d	2.26 ± 0.06
15	6.38	691.1888	691.1874	C_32_H_36_O_17_	2	94.75	Methyl 5-(6-caffeoyl-glucopyranosyl)-caffeoylquinic acid	1.67 ± 0.09
18	7.05	337.0909	337.0923	C_16_H_18_O_8_	−4.2	100	Coumaroylquinic acid isomer a	1.85 ± 0.10
19	7.28	337.0908	337.0923	C_16_H_18_O_8_	−4.4	99.80	Coumaroylquinic acid isomer b	2.75 ± 0.13
21	7.70	367.1023	367.1029	C_17_H_20_O_9_	−1.6	99.70	Feruloylquinic acid	<LOQ
24	8.33	551.1400	551.1401	C_25_H_28_O_14_	−0.2	99.86	Caffeoyl hexosyl trihydroxymethoxyphenyl propanoic acid	1.70 ± 0.10
27	9.00	705.1643	705.1651	C_32_H_34_O_18_	−2.3	98.26	Subulatin	<LOQ
43	11.81	411.1650	411.1655	C_20_H_28_O_9_	−1.2	99.94	Coumaric acid-malonyl-hexoside	7.35 ± 0.21
Flavonoids and derivatives								
7	5.00	305.0660	305.0661	C_15_H_14_O_7_	−0.3	99.70	Epigallocatechin	0.87 ± 0.08
26	8.64	531.1337	531.1339	C_25_H_24_O_13_	−0.4	99.98	6′′-*O*-Malonylglycitin	<LOQ
32	9.77	595.1298	595.1299	C_26_H_28_O_16_	−0.2	94.74	Quercetin3-*O*-arabinosylgalactoside	<LOQ
34	10.09	447.0934	447.0927	C_21_H_20_O_11_	1.6	96.17	Kaempferol 3-*O*-glucoside	<LOQ
35	10.13	609.1462	609.1456	C_27_H_30_O_16_	1	100	Quercetin-rutinoside isomer a *	<LOQ
36	10.29	609.1461	609.1456	C_27_H_30_O_16_	0.8	99.73	Quercetin-rutinoside isomer b *	3.59 ± 0.27
37	10.39	463.0885	463.0877	C_21_H_20_O_12_	1.7	99.84	Quercetin 3-*O*-galactoside isomer a	3.53 ± 0.36
38	10.53	463.0885	463.0877	C_21_H_20_O_12_	0.9	99.55	Quercetin 3-*O*-galactoside isomer b	3.73 ± 0.40
39	10.83	477.0645	477.0669	C_21_H_18_O_13_	5	99.99	Quercetin-3-glucuronide	7.27 ± 0.50
40	11.24	433.0751	433.0771	C_20_H_18_O_11_	−4.6	99.46	Quercetin-3-arabinoside	1.86 ± 0.13
41	11.41	505.0970	505.0982	C_23_H_22_O_13_	−2.4	99.95	Quercetin 3-(2″-acetylgalactoside) isomer a	4.37 ± 0.22
42	11.66	447.0921	447.0927	C_21_H_20_O_11_	−1.3	99.72	Quercetin-3-*O*-rhamnoside	<LOQ
44	11.90	505.0966	505.0982	C_23_H_22_O_13_	−3.2	80.48	Quercetin 3-(2″-acetylgalactoside) isomer b	0.28± 0.01
45	11.95	491.0803	491.0826	C_22_H_20_O_13_	−4.7	90.78	Isorhamnetin-glucuronide	0.21± 0.05
46	11.99	579.1350	579.1350	C_26_H_28_O_15_	0	95.15	Quercetin-3-*O*-R-arabinofuranoside	0.17 ± 0.01
47	12.15	505.0988	505.0982	C_23_H_22_O_13_	1.2	99.99	Quercetin 3-(2″-acetylgalactoside) isomer c	0.12 ± 0.01
48	12.49	489.1040	489.1033	C_23_H_22_O_12_	1.4	97.29	Kaempferol 3-*O*-acetyl-glucoside	0.89 ± 0.03
49	12.71	519.1134	519.1139	C_24_H_24_O_13_	−1	99.66	Isorhamnetin-acylated-hexoside	<LOQ
50	13.35	591.1363	591.1350	C_27_H_28_O_15_	2.2	99.98	Quercetin-HMG-rhamnoside	2.94 ± 0.04
51	15.09	329.0653	329.0661	C_17_H_14_O_7_	−2.4	99.78	3′.7-Dimethylquercetin	<LOQ
Condensed tannins								
16	6.72	577.1337	577.1346	C_30_H_26_O_12_	−1.6	99.98	Procyanidin dimer	2.51 ± 0.18
22	7.75	879.1778	879.1773	C_45_H_36_O_19_	0.6	89.69	Procyanidin-prodelphinidin trimer(1 A-type bond)	<LOQ
23	8.29	865.1976	865.1980	C_45_H_38_O_18_	−0.5	91.43	Procyanidin trimer	1.15 ± 0.09
25	8.57	863.1823	863.1825	C_45_H_36_O_18_	0.2	90.32	Procyanidin trimer (1A-type bond)	1.33 ± 0.10
28	9.11	739.1647	739.1663	C_39_H_32_O_15_	−2.2	93.04	Cinchonain II isomer a	<LOQ
29	9.35	739.1680	739.1663	C_39_H_32_O_15_	2.3	99.20	Cinchonain II isomer b	0.05 ± 0.06
30	9.43	451.1025	451.1029	C_24_H_20_O_9_	−0.9	99.97	Cinchonain I a	0.31 ± 0.12
33	9.86	577.1337	577.1346	C_30_H_26_O_12_	0.9	88.65	Procyanidin dimer	0.60 ± 0.06
Lignans								
17	6.95	553.1544	553.1557	C_25_H_30_O_14_	−2.4	98.82	Ligustrosidic acid	
20	7.56	553.1555	553.1557	C_25_H_30_O_14_	−0.4	99.80	Ligustrosidic acid	
Other compounds								
1	0.45–0.60	191.0545	191.0556	C_7_H_12_O_6_	−4.8	99.00	Quinic acid	
31	9.57	535.1448	535.1452	C_25_H_28_O_13_	−0.7	98.95	Coumaroyl iridoid (I)	
							Sum of phenolic acids	134.28 ± 2.73
							Sum of flavonoids	35.79 ± 2.11
							Sum of phenolic compounds	170.07 ± 4.84

* Identification confirmed by a commercial standard.

**Table 5 foods-12-01688-t005:** Minimum inhibitory concentration (MIC), minimum bactericidal concentration (MBC) and minimum fungicidal concentration (MFC) of bilberry leaves extract and sorbic acid (E-200) as positive control.

	Strain Type	Bilberry Leaves Extract		Sorbic Acid (E-200)	
		MIC * (mg/mL)	MBC/MFC * (mg/mL)	MIC * (mg/mL)	MBC/MFC * (mg/mL)
Gram-positive	*L. monocytogenes*	3.12	6.25	3.12	6.25
	*L. innocua*	3.12	6.25	3.12	6.25
	*S. aureus*	0.4	0.8	0.8	1.56
	*E. faecalis*	3.12	6.25	3.12	6.25
	*B. cereus*	0.4	0.8	1.56	3.12
Gram-negative	*S. enterica*	25	50	3.12	6.25
	*E. coli*	25	50	3.12	6.25
	*S. sonnei*	25	50	6.25	12.5
	*P. aeruginosa*	25	50	1.56	3.12
Fungi	*C. sake*	25	50	12.5	25
	*Z. bailii*	50	>50	6.25	12.5
	*P. expansum*	50	>50	6.25	12.5
	*A. niger*	50	>50	6.25	12.5

* Value was obtained from three independent experiments which showed identical results.

**Table 6 foods-12-01688-t006:** Antitumor activity of optimized bilberry leaves extract and positive control (5-fluorouracil). Results are expressed as the mean ± SD.

Cellular Line	IC_50_ (μg/mL)	
	Bilberry Leaves Extract	5-Fluorouracil
HT-29 (Human grade II colorectal adenocarcinoma)	213.2 ± 2.5	10.3 ± 0.2
T-84 (Human colorectal carcinoma)	1140.3 ± 5.2	27.0 ± 1.6
SW-837 (Human grade IV rectum adenocarcinoma)	936.5 ± 4.6	19.4 ± 0.7

## Data Availability

Data is contained within the article.

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
