# Peer review of "Establishment of a Sonotrode Extraction Method and Evaluation of the Antioxidant, Antimicrobial and Anticancer Potential of an Optimized Vaccinium myrtillus L. Leaves Extract as Functional Ingredient"

_foods, 2023, doi:10.3390/foods12081688_

Round 1

Reviewer 1 Report

The manuscript entitled “Establishment of a sonotrode extraction method and evaluation 2 of the antioxidant, antimicrobial and anticancer potential of an 3 optimized Vaccinium myrtillus L. leaves extract as functional 4 ingredient” by Gil Martinez et al. reports an innovative extraction method of Vaccinium myrtillus L. (bilberry) leaves set up through a BOX-Behnken design. The optimal extract was characterized by HPLC-MS identifying 53 compounds . Moreover, the extract was tested for the antimicrobial and anticancer properties reporting a good activities.

-       Please add the positive control for all the performed tests

-       In table 5 are reported 53 compounds recognized by HPLC-ESI-TOF-MS. Compounds have been identified only by comparison with different databases and literature. In my opinion this characterization needs to be consolidate by comparison with the standards, at least of the most representative compounds. 

-       In paragraphs 3.6 the antiproliferative activity of the sample have been reported. I think could be useful to insert in the table the positive control in order to better understand the data. The discussion reported includes all the analysis necessary to comprehend data but I think is useful to highlight it on the table.

Author Response

The manuscript entitled “Establishment of a sonotrode extraction method and evaluation 2 of the antioxidant, antimicrobial and anticancer potential of an 3 optimized Vaccinium myrtillus L. leaves extract as functional 4 ingredient” by Gil Martinez et al. reports an innovative extraction method of Vaccinium myrtillus L. (bilberry) leaves set up through a BOX-Behnken design. The optimal extract was characterized by HPLC-MS identifying 53 compounds . Moreover, the extract was tested for the antimicrobial and anticancer properties reporting a good activities.

The manuscript is well written, and suitable for publication in Food after revision of few points:

-       Please add the positive control for all the performed tests

Response: Authors thanks the reviewer’s opinion and the positive controls have been included for antimicrobial and antiproliferative assays. They were not added in the first version because we did not want to stablish a comparative between commercial substances or drugs, but with other extracts obtained in bibliography. Although we agree that it may be interesting for the reader to have this data as well.

-       In table 5 are reported 53 compounds recognized by HPLC-ESI-TOF-MS. Compounds have been identified only by comparison with different databases and literature. In my opinion this characterization needs to be consolidate by comparison with the standards, at least of the most representative compounds. 

Response: The authors appreciate the suggestion of the reviewer, and changes have been made in the manuscript to clarify this issue. It is true that in the manuscript it is not specifically written that the retention time and fragmentation have been compared with commercial standards but, as it can be seen in paragraph 2.1, chlorogenic acid (which was by far the most abundant molecule in the extract) was compared and quantified with a commercial standard, and also rutin. As it is known, phenolic compounds standards are very expensive and most of them are not commercially available. Because of that, the quantification of phenolic compounds is usually done using a standard from the same family of compounds (Foods 2023, 12(6), 1201; Foods 2023, 12(1), 192).

-       In paragraphs 3.6 the antiproliferative activity of the sample have been reported. I think could be useful to insert in the table the positive control in order to better understand the data. The discussion reported includes all the analysis necessary to comprehend data but I think is useful to highlight it on the table.

Response: The authors agree with the reviewer and the results obtained for the positive control (5-fluorouracil), have been added to the table.

Reviewer 2 Report

Lidia Gil-Martínez et al., In this manuscript entitled Establishment of a sonotrode extraction method and evaluation of the antioxidant, antimicrobial and anticancer potential of an optimized Vaccinium myrtillus L. leaves extract as functional ingredient. I think it’s better to discuss about below questions. Therefore, I suggest a major revision for the manuscript before publication.

-The manuscript is written well, but grammar errors should be done extensively?

Abstract: A good abstract must have the followings: Important results and results that supported them. Concluding remark.

- The abbreviations mentioned in the text. Ex…page 2 ; line 84.. DPPH, ABTS, TPTZ, potassium persulfate, gallic acid and MBC/MFC in table 6.

Material methods:

-Authors should give details of vendors of the chemicals used.

-Report the statistical analysis in a dedicated section and the significance of results; All The statistics information such as (Mean ± SEM and the statistical test used to calculate a p-value should be indicated under the Table 4.

-Author should calculate the minimum inhibitory concentration against pathogenic microbes to check the effectiveness of Vaccinium myrtillus L. leaves

-Why the natural product is having antioxidant and antimicrobial activity, that to be discussed in the introduction section by citing the following articles: doi.org/10.1007/s12010-022-04241-8 and doi.org/10.1007/s12257-021-0211-1

-In vitro antiproliferative assay. The authors should be explaining how choice concentrate inhibitory concentration (IC50) values of the extract for the cancer cells.

-The authors should be explaining about the importance and novelty of the work in more detail.

Author Response

Lidia Gil-Martínez et al., In this manuscript entitled Establishment of a sonotrode extraction method and evaluation of the antioxidant, antimicrobial and anticancer potential of an optimized Vaccinium myrtillus L. leaves extract as functional ingredient. I think it’s better to discuss about below questions. Therefore, I suggest a major revision for the manuscript before publication.

-The manuscript is written well, but grammar errors should be done extensively?

Response: A comprehensive review has been made through the text in order to correct the grammar mistakes.

Abstract: A good abstract must have the followings: Important results and results that supported them. Concluding remark.

Response: The authors thanks the reviewer for the suggestion and for this reason, changes have been made in the abstract with the intention to improve it.

- The abbreviations mentioned in the text. Ex…page 2 ; line 84.. DPPH, ABTS, TPTZ, potassium persulfate, gallic acid and MBC/MFC in table 6.

Response: The comment of the reviewer was appreciated by the authors and changes have been made through the manuscript in order to correct the writing mistakes.

Material methods:

-Authors should give details of vendors of the chemicals used.

Response: Vendors of the chemicals used are specified in epigraph 2.1.

-Report the statistical analysis in a dedicated section and the significance of results; All The statistics information such as (Mean ± SEM and the statistical test used to calculate a p-value should be indicated under the Table 4.

Response: Authors appreciate the suggestion of the reviewer and a new section has been dedicated to statistic analysis epigraph 2.8. Furthermore, table 4 and the rest of tables with results (5 and 6) have been improved adding the statistical information.

-Author should calculate the minimum inhibitory concentration against pathogenic microbes to check the effectiveness of Vaccinium myrtillus L. leaves.

Response: The authors welcome the comment of the reviewer and, as it was possible to repeat the assay, MIC values against pathogenic microbes have been calculated.

-Why the natural product is having antioxidant and antimicrobial activity, that to be discussed in the introduction section by citing the following articles: doi.org/10.1007/s12010-022-04241-8 and doi.org/10.1007/s12257-021-0211-1

The authors thanks the reviewer for the advice and have enjoyed reading the papers suggested. As the natural product is having antioxidant activity and antimicrobial activity thanks to its content in phenolic compounds and their molecular structure (as mentioned in the introduction, both articles have been cited in the manuscript ([8] and [13]) to support the information given.

-In vitro antiproliferative assay. The authors should be explaining how choice concentrate inhibitory concentration (IC50) values of the extract for the cancer cells.

Response: The calculation of IC50 values is described in methodology section 2.7.2.

-The authors should be explaining about the importance and novelty of the work in more detail.

Response: Authors thank and appreciate the suggestion of the reviewer and changes have been made in the manuscript in order to clarify the novelty and importance of this research.

Reviewer 3 Report

Comments on this manuscript

In this manuscript, the authors use an ultrasound-assisted method to extract polyphenols from the leaves and investigate the effect of these substances on the inhibition of microorganisms and cancer model cells. However, ultrasound power, frequency, time and intermittency of operation all affect the effectiveness of the extraction of bioactive compounds; temperature and extraction solvent are also important factors affecting extraction. Why did the authors only consider "ethanol % (v/v) (X1), extraction time (min) (X2) and amplitude (%) (X3)" and not temperature?

Plant material should provide more detailed information, such as harvesting date, drying method and moisture content, etc.

The extract has an inhibitory effect on cancer cells in in vitro experiments, which in practice is not the same as necessarily having anti-cancer activity in vivo. The novelty of this work is not clear.

Author Response

In this manuscript, the authors use an ultrasound-assisted method to extract polyphenols from the leaves and investigate the effect of these substances on the inhibition of microorganisms and cancer model cells. However, ultrasound power, frequency, time and intermittency of operation all affect the effectiveness of the extraction of bioactive compounds; temperature and extraction solvent are also important factors affecting extraction. Why did the authors only consider "ethanol % (v/v) (X1), extraction time (min) (X2) and amplitude (%) (X3)" and not temperature?

Response: Temperature was not stablished as a factor to optimize because, amplitude and extraction time are directly correlated with temperature. The highest the amplitude and the longest the time of extraction, the highest the temperature. Thus, in the model of this study, temperature has been optimized from an indirect point of view by optimizing time of extraction and amplitude. Furthermore, from an industrial point of view, controlling temperature during ultrasonic assisted extraction is more difficult, energy consuming and expensive than controlling the amplitude and time of extraction.

Plant material should provide more detailed information, such as harvesting date, drying method and moisture content, etc.

Response: Authors thank and appreciate the suggestion of the reviewer and the information required has been added to the manuscript.

  • The extract has an inhibitory effect on cancer cells in in vitro experiments, which in practice is not the same as necessarily having anti-cancer activity in vivo.

Response: The authors welcome the appreciation of the reviewer and agree with him. In order to avoid misunderstanding, changes have been made in the manuscript. Future research lines would be focused on the deeper investigation of anticancer activity potential of bilberry leaves with more in vitro and in vivo assays. For the moment, IC50 for tumoral cell lines is considered a valid method to test and do an screening of the potential of a molecule or a extract to have anticancer activity.

The novelty of this work is not clear.

Response: Authors appreciate the opinion of the reviewer and in order to clarify the novelty of the work, changes have been made in the manuscript. To our knowledge, this is the first work published of a sonotrode assisted extraction method for the obtention of bioactive compounds from bilberry leaves. Furthermore, there are 22 new compounds that have been characterized for the first time in bilberry leaves and the antitumor potential of bilberry leaves has been also evaluated for the first time. This research may be the precursor of a new line of research focused on the evaluation of the potential of phytochemicals from bilberry leaves such as the antioxidant or antitumor activity in order to develop nutraceuticals or functional foods that may be helpful in the prevention and treatment of different diseases.